# Exploring the Interplay of Food Security, Safety, and Psychological Wellness in the COVID-19 Era: Managing Strategies for Resilience and Adaptation

**DOI:** 10.3390/foods13111610

**Published:** 2024-05-22

**Authors:** Fanrui Zhou, Zhengxin Ma, Ahmed K. Rashwan, Muhammad Bilawal Khaskheli, Wessam A. Abdelrady, Nesma S. Abdelaty, Syed Muhammad Hassan Askri, Ping Zhao, Wei Chen, Imran Haider Shamsi

**Affiliations:** 1Department of Food Science and Nutrition, College of Biosystems Engineering and Food Science, Zhejiang University, Hangzhou 310058, China; 2Key Laboratory of State Forestry and Grassland Administration on Highly Efficient Utilization of Forestry Biomass Resources in Southwest China, College of Material and Chemical Engineering, Southwest Forestry University, Kunming 650224, China; 3Zhejiang Key Laboratory of Crop Germplasm Resource, Department of Agronomy, College of Agriculture and Biotechnology, Zhejiang University, Hangzhou 310058, China; 4Department of Food and Dairy Sciences, Faculty of Agriculture, South Valley University, Qena 83523, Egypt; 5Law School, Dalian Maritime University, Dalian 116026, China; 6Department of Agronomy, Faculty of Agriculture, South Valley University, Qena 83523, Egypt; 7Department of Dairy Science, Faculty of Agriculture, Kafrelsheikh University, Kafr El-Sheikh 33516, Egypt; 8Key Laboratory for Forest Resources Conservation and Utilization in the Southwest Mountains of China, Ministry of Education, Southwest Forestry University, Kunming 650224, China

**Keywords:** food safety, COVID-19, eating disorders, food security, psychological health

## Abstract

The global population surge presents a dual challenge and opportunity in the realms of food consumption, safety, and mental well-being. This necessitates a projected 70% increase in food production to meet growing demands. Amid this backdrop, the ongoing COVID-19 pandemic exacerbates these issues, underscoring the need for a deeper understanding of the intricate interplay between food consumption patterns and mental health dynamics during this crisis. Mitigating the spread of COVID-19 hinges upon rigorous adherence to personal hygiene practices and heightened disease awareness. Furthermore, maintaining stringent food quality and safety standards across both public and private sectors is imperative for safeguarding public health and containing viral transmission. Drawing upon existing research, this study delves into the pandemic’s impact on mental health, food consumption habits, and food safety protocols. Through a comprehensive analysis, it aims to elucidate the nuanced relationship among food, food safety, and mental well-being amid the COVID-19 pandemic, highlighting synergistic effects and dynamics that underpin holistic human welfare. Our study offers a novel approach by integrating psychological wellness with food security and safety. In conceiving this review, we aimed to comprehensively explore the intricate interplay among food security, safety, and psychological wellness amid the backdrop of the COVID-19 pandemic. Our review is structured to encompass a thorough examination of existing research, synthesizing insights into the multifaceted relationships among food consumption patterns, mental health dynamics, and food safety protocols during the crisis. Our findings provide valuable insights and practical recommendations for enhancing food security and psychological well-being, thus supporting both academic research and real-world applications in crisis management and policy development.

## 1. Introduction

The COVID-19 outbreak was declared a pandemic by the World Health Organization (WHO) in March 2020; the global spread of the pandemic has deeply impacted millions of individuals, resulting in more than 400 million confirmed cases and the unfortunate loss of 6 million lives worldwide. The far-reaching consequences of this phenomenon have been devastating [1]. Several national and municipal administrations implemented preventive health measures, including stringent restrictions like mask requirements, quarantines, lockdowns, and closures of non-essential services to control the global spread of the coronavirus that causes severe acute respiratory syndrome (SARS-CoV-2). These actions had significant sociocultural and economic repercussions. Food advisories have played a pivotal role amid the COVID-19 pandemic, intertwining with multiple facets such as public health, food safety, consumer behavior, and regulatory protocols. Primarily, these advisories have been instrumental in imparting precise information to the public concerning safe food handling practices, potential hazards linked with food consumption, and strategies to mitigate virus transmission in food-related endeavors. The epidemic has brought about disruptive and stressful societal changes that have an impact on the general public’s mental health. Additionally, infected people develop immediate and long-lasting neuropsychiatric symptoms, either as a result of the initial infection or as a result of the post-acute COVID syndrome [2], which refers to symptoms lasting over 3–4 weeks and affecting multiple organs, including the brain [3]. During the COVID-19 pandemic, individuals worldwide have faced a multitude of stressors that have profoundly impacted mental health. The fear of contracting the virus and worries about loved ones’ health have led to heightened anxiety and stress [4]. Social isolation measures, such as lockdowns and restrictions on gatherings, have exacerbated feelings of loneliness and disconnection [5]. Economic instability, job losses, and financial worries have also contributed to heightened stress levels and uncertainty about the future [6]. The disruption of daily routines, constant exposure to news updates, and misinformation have further fueled feelings of overwhelm and confusion [7]. Moreover, the loss of loved ones due to COVID-19, coupled with the inability to mourn or attend funerals, has led to complicated grief and emotional distress [8]. Concerns about overwhelming healthcare systems and limited access to medical care have added to individuals’ stress and anxiety, particularly for those with pre-existing health conditions [9].

Increased food supply is necessary to meet the problems posed by the epidemic and the anticipated global population of 9 billion people by 2030. Agricultural and animal production must be industrialized and intensified to enable sustainable innovation. In addition, there is an increasing demand for food safety management, preparation, and storage to lower the risk of food-borne infections and food waste [10]. To overcome global difficulties, it is essential to improve nutrition, human health, food safety, and food cleanliness through research and wise decision-making. Practical, evidence-based policies should consider the connection between food and mental health. However, differences in organizational levels, infrastructure, educational potential and food protection norms among nations in the larger international environment pose difficulties [11]. Raising awareness about food safety and establishing robust government laws and enforcement are essential to address these issues [12]. In response to the multifaceted challenges posed by the COVID-19 pandemic, our study offers a novel approach by integrating psychological wellness with food security and safety. In conceiving this review, we aimed to comprehensively explore the intricate interplay among food security, safety, and psychological wellness amid the backdrop of the COVID-19 pandemic. Our review is structured to encompass a thorough examination of existing research, synthesizing insights into the multifaceted relationships among food consumption patterns, mental health dynamics, and food safety protocols during the crisis. Unlike previous research that often addresses these domains individually, our manuscript provides a comprehensive framework that examines the synergistic effects of pandemic-induced stressors on food consumption patterns and safety protocols. To ensure the rigor and relevance of our analysis, we adopted inclusion criteria focused on studies and reports that specifically addressed the impacts of COVID-19 on global mental health, food consumption behaviors, and food safety practices. Additionally, we prioritized research that offered insights into strategies for resilience and adaptation in the face of these challenges. By proposing actionable strategies for resilience and adaptation, this study aims to enhance both individual and community well-being, filling a critical gap in the current literature and offering practical solutions for policymakers and stakeholders globally.

## 2. Navigating Food Safety and Industry Challenges in the Era of COVID-19

With the WHO declaring severe acute respiratory syndrome coronavirus 2 (SARS-CoV-2) a global health emergency in March 2020, various aspects of life have been met with both risks and opportunities [13]. Referred to as the new coronavirus causing COVID-19, SARS-CoV-2 shares specific characteristics with influenza [14]. Despite differing transmission modes, both viruses lead to respiratory diseases. Apart from common spread routes like human contact and droplets, SARS-CoV-2 also transmits via aerosols, especially indoors [15,16]. This means infected individuals can spread the virus through activities like handshakes, handling contaminated objects, coughing, or sneezing, thereby producing virus-laden droplets that can become aerosols [17]. Consequently, many governments have enforced strict isolation measures, affecting the food industry. This includes enforcing control systems, overseeing compliance with European Commission standards domestically and abroad, managing science-based risk management with the European Food Safety Authority (EFSA) [18,19,20]. The decision tree for food safety, as per EFSA regulation, is shown in Figure 1.

### 2.1. Food Consumption Amid the COVID-19 Pandemic

Food consumption during the COVID-19 pandemic has been influenced by changes in daily routines, social connections, and psychological well-being [21]. Isolation and remote work disrupted regular activities, resulting in increased sedentary behavior and fewer opportunities for physical exercise [22,23]. This shift led to more frequent meals and snacks [22]. Additionally, the pandemic’s impact on mood and emotional well-being influenced food choices [24]. While some experienced improved nutrition quality during lockdown, others faced deterioration or no significant changes [25]. The pandemic’s aftermath has led to substantial lifestyle changes, affecting body weight, food choices and physical activity [26,27]. The increased consumption of unhealthy snacks and reduced physical activity resulted in weight gain among adults during quarantine [28,29]. Adverse changes in weight-related eating behaviors and perceived barriers to weight management were also reported in the UK [30]. The pandemic also saw a rise in convenience food consumption, whereas visits to markets and restaurants declined and home-cooked meals became more common [27]. Consumers’ patterns varied across countries, with some opting for healthier and higher-quality foods, while others preferred unhealthy ones [31,32,33]. The pandemic also influenced food waste, reducing it due to changed consumption behaviors [34,35]. Emotional factors played a significant role, with stress and negative emotions often leading to emotional eating [36,37,38].

### 2.2. Addressing Food Safety Challenges in the Pandemic

The COVID-19 pandemic severely disrupted food production and supply chains due to labor shortages, transportation constraints, and the closure of food services, including restaurants [39]. In the face of the ongoing pandemic, it is of utmost importance to prioritize and ensure food safety. This is vital for effectively managing the risks associated with food-borne diseases and maintaining a dependable, safe, healthy, and nutritious food supply. Food producers and handlers have assumed greater responsibility in implementing robust measures to safeguard food safety, which includes conducting thorough risk analyses and implementing strategies to combat pathogen resistance (Figure 2). In developing countries, the prevalence of public health hazards linked to food safety tends to be higher compared to developed countries. This can be partly attributed to inadequate sanitation and hygiene practices [40]. Moreover, weak regulatory services in these countries often contribute to non-compliance with food hygiene and safety regulations, leading to increased risks of food contamination [41]. Additionally, it is notable that countries at different stages of development may have disparate levels of legislation governing acceptable levels of harmful contaminants in food.

### 2.3. The Resilience of the Food Industry during the Pandemic

The COVID-19 pandemic has profoundly impacted the food industry, presenting challenges related to food security, safety, sustainability, and marketing [42]. Ensuring food security involves protecting the national food production chain from the transmission and persistence of contagious and chemical agents [43]. Disruptions in food supply chains have increased the risks to food safety, especially for perishable foods that require efficient distribution to maintain freshness [44]. Concerns about the virus spreading through aerosols have highlighted the potential for sick workers to contaminate food and equipment [45]. There are also concerns regarding virus-laden dewdrops on surfaces such as equipment and packaging, which may pose additional risks [46]. SARS-CoV-2 has demonstrated the capability to survive on various surfaces, including plastics and stainless steel, for extended periods, potentially leading to contamination [47]. In crowded environments, such as supermarkets and wholesale markets, the virus can spread rapidly through droplets or aerosols [48]. Furthermore, the frozen food supply chain presents transmission risks, as the virus can remain infectious during transportation and maintain its viability for several days under refrigeration. [49,50]. Implementing risk management strategies becomes crucial to mitigate contamination risk, particularly in the case of cold-chain products [51,52]. Consumer behaviors like hoarding and panic buying, driven by misinformation, have further impacted the stability and efficiency of the food supply chain [53].

In response, the food industry has adapted by implementing fixed-point deliveries to minimize human contact and contamination risk [54]. Online meal deliveries have gained significant popularity for reducing interpersonal interaction and physical separation between buyers and suppliers [54]. Establishing comprehensive food safety traceability system and adhering to proper hygiene habits and sanitation practices have become crucial in ensuring food safety [55,56]. Such measures enable the effective monitoring and tracking of food products throughout the supply chain, ultimately safeguarding the integrity and safety of the food supply.

## 3. Exploring the Intersection of Food and Mental Health

Food and nutrition are pivotal in shaping both physical and mental health. The absence of essential nutrients can significantly impact mental well-being, contributing to conditions like anxiety and depression. Conversely, improving one’s diet has shown promise in alleviating these mental health challenges and promoting overall well-being [57,58,59,60,61,62,63,64,65,66,67]. Recent studies underscore the importance of nutritional deficiencies in influencing physical and mental health.

### 3.1. The Dynamic Relationship between Food and Mental Health

The intricate relationship between food and mood, coupled with the reciprocal influence of mental state on food choices, warrants elucidation. Emotions influence food consumption in a dynamic loop, which subsequently affects mood and shapes future dietary decisions. Understanding this complex interplay is vital for effectively managing mental health [68]. Specific nutrients play a crucial role in regulating cognitive processes and emotions. Conversely, deficiencies in essential vitamins, fatty acids, minerals, and macronutrients, notably proteins, can exacerbate negative mental states [68]. Recognizing the impacts of nutrients on mental well-being underscores the significance of maintaining a balanced diet to support overall cognitive health (Figure 2).

### 3.2. Impact of Nutrients on Mental Well-Being

Diets abundant in carbohydrates elicit the biosynthesis of critical brain chemicals, including 5-HT and TRP, which have a significant impact on mood and behavior. Consuming low glycemic index (GI) foods, such as fruits and vegetables and complex carbohydrates like whole grains and pasta, exert a prolonged and more potent influence on brain chemistry, mood regulation, and sustained energy levels than sugar consumption [63]. Furthermore, the intake of proteins exerts profound effects on brain functioning and mental health due to their intricate array of amino acids. Dopamine, derived from tyrosine, and serotonin, sourced from tryptophan, play pivotal roles in mood modulation. Insufficient levels of these crucial amino acids may lead to a dearth of neurotransmitter synthesis and subsequent low mood, while excesses might trigger detrimental brain impairments and cognitive deficits [63]. Research indicates that diminishing tryptophan levels tend to disturb mood more profoundly than increasing carbohydrate intake, underscoring the critical role of these amino acids in mental well-being [62,69]. Additionally, the brain is considered a lipid-rich organ, predominantly comprising phospholipids, sphingolipids, and cholesterol in the brain membrane. Remarkably, 50% of these lipids consist of polyunsaturated fatty acids (PUFAs), with the omega-3 family representing a significant proportion. Clinical and epidemiological evidence underscores the pivotal role of n-3 PUFA in mental health, and deficiencies in the diet have been associated with heightened risks of diverse mental disorders, including depression [65]. In addition, other micronutrients play an essential role in mental well-being; for example, folate insufficiency, commonly observed as depression, marks one of the most prevalent signs of micronutrient deficiency. Depression patients often exhibit 25% lower blood folate (B9) levels compared to their healthy counterparts. Vitamins B6 and B12 play indispensable roles in producing various neurotransmitters, and cobalamin (B12) supplementation has been shown to enhance cerebral and cognitive capabilities while preserving myelin sheath integrity [63]. Remarkably, long-term vitamin intake, particularly B1, B2, and B6, has been associated with significant mood enhancements, as shown in Figure 2 [59]. Moreover, emerging research suggests that certain dietary supplements, such as nano-curcumin, may also play a role in promoting mental well-being by modulating inflammatory responses in the body. For instance, treatment with nano-curcumin has been shown to significantly reduce IL-1 and IL-6 gene expression and secretion, which are key markers of inflammation, in the serum and supernatant of individuals affected by COVID-19 [70,71]. While the direct link between nano-curcumin and mental health requires further exploration, these findings highlight the intricate connections among nutrition, inflammation, and mental well-being.

### 3.3. Impact of Nutrients on Immune System

The availability of the COVID-19 vaccine notwithstanding, the importance of dietary habits in maintaining overall well-being and bolstering resistance cannot be overstated [72]. Critical micronutrients such as vitamins C, A, E, and D, iron, zinc, folic acid, probiotics, and prebiotics play significant roles in supporting our immune systems to varying degrees [73,74,75,76,77,78,79,80]. These nutrients are abundantly found in fruits, vegetables, herbs, seeds, nuts, and cereals, as well as superfoods like chlorella and spirulina. Additionally, fortified foods and functional foods rich in bioactive compounds, along with plant-based foods, have demonstrated significant potential in enhancing immunity against viral infections such as COVID-19 [81]. Several clinical trials and retrospective cohort studies have indicated that a plant-based diet, along with supplementation of vitamins D and C, probiotics, and zinc salts, may reduce the severity of COVID-19 symptoms. These critical nutrients derived from daily dietary intake play a pivotal role in enhancing the immune system (Figure 3). For example, vitamin D has been shown to modulate both the innate and adaptive immune systems [82]. Within the innate system, toll-like receptors (TLR) aid macrophages in identifying lipopolysaccharide (LPS), a bacterial immunity surrogate. Subsequently, peptides with potent bactericidal properties, such as β-defensin 4 and cathelicidin, are generated following TLR engagement, leading to the disruption of the bacterial cell membrane and subsequent antimicrobial activity within the phagosomes [83]. Supplementation with vitamin C can facilitate recovery from acute respiratory infections by acting as a cofactor for gene regulatory and biosynthetic enzymes. Moreover, it enhances processes such as phagocytosis, neutrophil migration, oxidant generation, and microbial elimination [84]. The mineral zinc is crucial for the growth and function of immune cells, playing a vital role in protecting against pathogens while also regulating an overactive immune response. However, both inadequate and excessive zinc intake can impair its efficacy. By optimizing immune function and providing anti-inflammatory support, zinc acts as a guardian of immune health [85,86]. In conclusion, a balanced diet comprising fruits, vegetables, high-quality protein, and whole-grain foods is essential. Dietary guidance from nutritionists and healthcare professionals is advisable, and the trend toward personalized nutrition is gaining momentum.

### 3.4. Insights into Mental Health Disorders in the Modern World

Mental disorders, encompassing a spectrum of conditions like depression, anxiety, bipolar disorder, eating disorders, schizophrenia, substance abuse disorders, and neurodevelopmental conditions like autism, attention-deficit hyperactivity disorder (ADHD), and developmental disabilities, manifest through aberrant cognitive processes, emotional responses, behavioral patterns, and interpersonal interactions. Various factors, including coping mechanisms, social dynamics, environmental stressors, occupational conditions, and community support, shape mental health outcomes. Understanding the multifaceted nature of mental health disorders is crucial for developing effective interventions and support systems.

### 3.5. Influence of Nutrition and Hunger on Mental Health

The brain, akin to other organs, relies on vital nutrients from food to fuel its activities, significantly influencing mood and emotions. Food affects mood through gradual changes in brain chemistry and sensory stimulation, encompassing taste, aroma, and flavor [68,69]. Notably, certain foods, like coffee and chocolate, have direct links to mood enhancement. Even a small amount of chocolate is believed to elevate happiness and improve mood, while coffee provides a sense of energy and alertness. Caffeine, theobromine, and sweet taste, coupled with psychological mechanisms, contribute to mood improvement, and chocolate, when consumed in adequate amounts on an empty stomach, may promote serotonin synthesis [68,69]. Serotonin (5-HT and TRP), synthesized from the essential amino acid tryptophan, has long been associated with sleep and affective disorders, such as sadness and anxiety, and has demonstrated mood-enhancing effects [68,69,87].

Conversely, hunger induces emotional fluctuations in mood, perceptions, and behavior due to changes in glucose levels, leading to the release of specific hormones that influence emotional states. The theory connecting mood changes to blood sugar levels underscores the significance of maintaining adequate glucose levels for emotional regulation [57]. Decreases in glucose levels can result in impulsivity, irritability, and heightened aggression, triggering the release of hormones like cortisol and adrenaline to restore balance. Neuropeptide Y, another hormone, stimulates hunger sensations when the body requires additional nourishment, contributing to irritability and aggressive behavior [57].

### 3.6. Mental Health Shapes Food Choices

Various research findings indicate that individuals eat emotionally to respond to positive and negative emotional states. Notably, when experiencing unhappiness, emotional eaters exhibit a significant increase in food consumption compared to moments of joy. Additionally, during emotional eating episodes, there is a clear preference for sweet foods over salty ones [88]. Furthermore, the influence of mood on food choices has been extensively studied, revealing intriguing patterns. Positive moods steer individuals toward healthier food choices, aligning with long-term health objectives. On the other hand, negative moods trigger a desire to indulge in comfort foods, which serve to address immediate emotional needs [89]. Researchers have also delved into the psychological and physiological underpinnings of food preferences. Certain meals have become closely associated with offering relief from distress and altering emotional states or moods. This phenomenon highlights the strong impact of mood on food selection, with comfort foods being sought during specific circumstances, such as feelings of illness, sadness, or loneliness [90,91]. Moreover, gender differences play a role in comfort food choices. Males tend to gravitate toward hearty, warming comfort meals like steak, casseroles, and soup, while females prefer comfort foods such as chocolate and ice cream [68,92]. To explain the mood-enhancing effects of certain foods, research has revealed that high-calorie sweet treats like ice cream, cookies, and chocolate trigger an increase in serotonin and opiate production, thus contributing to mood elevation [93].

### 3.7. COVID-19’s Influence on Mental Health and Dietary Habits

The COVID-19 pandemic has triggered a global surge in mental health problems, including anxiety, depression, and post-traumatic stress disorder, with adverse effects on mental health services worldwide (Figure 4) [94]. Factors such as lockdowns, social isolation, loss of income, and restricted access to essential services have contributed to heightened distress and anxiety among individuals [95]. Emotional eating has become prevalent, with people consuming comfort foods like sugary snacks and sweet beverages as a coping mechanism during the pandemic [38]. Increased sedentary behavior and unhealthy eating habits have also led to weight gain and obesity, which, in turn, have been associated with worsened COVID-19 outcomes [96,97,98]. Chronic inflammation caused by poor diets further impairs the body’s defense against viral infections [99]. Moreover, mental health issues arising from the pandemic have added to the burden, with healthcare professionals facing significant emotional distress and the risk of suicide [11,43].

Financial stability and resource access have been associated with better mental health outcomes during the pandemic [100]. However, environmental stresses and social isolation have affected dietary patterns, leading to low intakes of essential nutrients among isolated individuals [101]. This decline in healthy food choices and reduced physical activity may have long-term implications for obesity and non-communicable diseases [102]. Beyond the pandemic, mental illnesses are already a significant global health issue [103,104,105]. Environmental stresses, including those brought on by the pandemic, can exacerbate mental health issues, but some individuals have experienced positive changes, such as improved family relations and work–life balance [106]. The pandemic’s impact on mental health and dietary behavior underscores the need for comprehensive mental health support and public health interventions to address these challenges.

## 4. Ensuring Food Security and Safety Amid the COVID-19 Crisis

It is believed that a significant issue for local governments is to produce favorable food hygiene environments for street foods, providing infrastructure, training staff, and offering building capacity to provide the “best available expertise”. Additionally, rigorous risk assessment strategies must be implemented across all stages of food production to uphold regulatory compliance and meet consumer expectations [107]. Healthcare professionals also play a crucial role in recognizing and treating food-borne illnesses through clinical supervision and dietary interventions (Figure 5).

### 4.1. Big Data Impacting Dynamic Food Safety in the Food Chain

Ensuring safe food production requires a risk-based approach that combines HACCP with prerequisite programs. To achieve a more precise quantitative risk management approach, the International Commission on Microbiological Specifications for Foods (ICMSF) recommends using risk management metrics such as food safety objectives (FSO) and performance objectives (PO) [108,109]. These metrics have been adopted by the Codex Alimentarius Commission and can be utilized by intergovernmental agencies, national governments, and food businesses. To make informed real-time food safety decisions, all stakeholders, including risk assessors, managers, and communicators, need access to food safety-related data collected throughout the food chain.

The emergence of the internet of things (IoT) and the digital revolution have resulted in rapidly collecting and transferring vast amounts of data in real time. This phenomenon, commonly known as “big data”, can significantly affect numerous aspects of the food industry, such as food safety, public health, and trade. The primary defining features of big data are its size, speed, accuracy, and diversity, which ultimately determine its worth in generating valuable insights and knowledge [110].The advent of smart farming, which involves the digitalization of farming practices [111,112], has led to the emergence of precision agriculture and “omics-based” precision food safety [109]. This innovative approach employs data to enhance foodborne outbreak traceability, predictive analytics, AI, and machine learning in food safety management. The digital revolution in food manufacturing and the supply chain also makes use of data to improve risk management, real-time transparency, and trend analysis. In order to uphold the standards of food safety and quality, it is imperative to extract valuable insights from data while ensuring seamless interoperability across various data sources. Although data scientists can easily create tools and dashboards to visualize data, the availability of precise and dependable metadata in a user-friendly format that domain experts can easily interpret is vital. This is a challenge that lies ahead in the realm of food safety and quality [113].

### 4.2. Innovations in Food Quality Assurance Technologies

To guarantee hygienic and nutritionally beneficial food for human health, implementing new legislation accompanied by technological advancements is imperative [114]. One such advancement is utilizing high-performance omics techniques in food quality detection, encompassing genomics, transcriptomics, proteomics, and metabolomics. Particularly, whole genome sequencing (WGS) has revolutionized food safety by providing intricate genetic insights into pathogens and spoilage organisms (Figure 6). WGS facilitates rapid outbreak detection, precise source tracking, and targeted interventions, thus significantly enhancing food quality detection. Its widespread adoption by public health authorities and the food industry underscores its vital role in ensuring the production of safer food products [115]. The operational mechanism of WGS, as elucidated by scientific research [116], encompasses the following:Omics techniques (genomics, transcriptomics, proteomics, and metabolomics) scrutinize DNA, mRNA, and protein alterations under diverse physiological and environmental conditions.The adoption of omics techniques broadens sampling programs, amplifying the detection of both known and unknown microorganisms along the food chain, including spoilage and pathogenic agents, thus enhancing food safety.WGS, a pivotal omics technique, facilitates the early identification of foodborne illness outbreaks and traces microorganisms throughout the food production continuum. High-resolution subtyping aids in pinpointing outbreak origins and contamination incidents.Omics approaches offer insights into microbial adaptation along the food continuum and strain-specific characteristics impacting human and animal health (e.g., virulence genes and antimicrobial resistance), guiding preventive measures and improving food safety.Integrating omics with other disciplines (e.g., genomics, proteomics, and metabolomics) reveals intricate interactions among phenomes, genomes, and environmental factors, enhancing the comprehension of crop breeding strategies and refining crop management practices.

Moreover, WGS, as a potent technology, holds immense potential in augmenting food quality detection [87,88,89]. Its mechanisms entail the following:Revealing genetic information: WGS provides a comprehensive view of an organism’s DNA, allowing for a detailed analysis of its genetic makeup. It offers unmatched precision in understanding genetic variations within and between species.Widely adopted: WGS has been widely embraced by public health, food safety authorities, and the food industry for various food safety and quality applications.Tracing transmission chains: WGS is used to trace pathogens and spoilage organisms’ introduction and transmission chains in food production. It aids in identifying and assessing microbiological quality issues, guiding mitigation efforts for safer food production.Predicting adaptation and survival: WGS predicts the adaptation and survival of pathogens and spoilage organisms, including their resistance to biocides and metals. This information helps prevent their persistence in the production environment.Outbreak investigation: WGS aids in outbreak investigations by clearing food products and identifying the outbreak profile. It can rule out producers if the profile is not found, aiding in swift intervention and prevention.Monitoring and persistence: The monitoring of emerging pathogens and antimicrobial resistance aids in early detection and control.Source identification: WGS identifies the source of contamination for a food product, enabling public health officials to locate and remove harmful ingredients quickly. This facilitates rapid response and prevents additional illnesses.Comprehensive information: WGS provides valuable insights into pathogenicity, virulence, adaptation, survival, and resistance, allowing for the design and implementation of effective interventions.Improving safety and quality: The enhancement of the safety, quality, and shelf life of foods can be achieved through understanding bacterial genes associated with public health and spoilage.

While WGS has been widely implemented, challenges such as implementation costs and uncertainties remain, necessitating ongoing research and development efforts.

### 4.3. Mitigating COVID-19 Transmission Risks in Food Handling

In the backdrop of the COVID-19 pandemic, both the Food and Drug Administration (FDA) and the WHO have issued recommendations aimed at bolstering food safety protocols [117]. These guidelines emphasize the meticulous utilization of masks, goggles, and gloves, along with frequent handwashing and the diligent sanitization of all surfaces involved in food preparation. Additionally, ensuring physical distancing is paramount, and the deployment of plexiglass barriers can effectively prevent food contact. It is imperative to thoroughly disinfect all surfaces and cultivate a culture of personal hygiene, particularly for individuals operating near food service areas [118,119]. To maintain a secure environment at retail establishments, prudent measures such as restricting customer footfall, actively enforcing physical distancing, and providing comprehensive sanitation provisions are critical. Adhering to safety protocols during food delivery, including contactless delivery and the meticulous handling of transportation containers, can significantly mitigate infection risks. Sound agricultural practices are also essential for fostering safe and wholesome food production [120]. In pursuit of food safety objectives, the four phases of the Food Safety Objectives/Appropriate Level of Protection system play a crucial role [121]. These phases aim to achieve optimal human and animal health, ensure safe agricultural practices, enhance working conditions, and raise awareness regarding common foodborne illnesses [122]. The WHO collaborates closely with governments to combat public health risks stemming from unsafe food and endeavors to safeguard consumer interests in securing a safe food supply [123]. The prudent allocation of resources and the implementation of educational initiatives have the potential to enhance national food systems and encourage mindful food management practices [124]. During the COVID-19 epidemic, ensuring food safety at restaurants has emerged as an urgent priority. Educating staff members on rigorous food safety protocols and encouraging participation in certification programs can significantly bolster consumer confidence.

### 4.4. Tactics for Preserving Food Safety and Mental Well-Being throughout the Pandemic

Amid the COVID-19 pandemic, the paramount importance of both food safety and mental well-being cannot be overstated. The implementation of stringent isolation measures has significantly impacted dietary habits, underscoring the urgent need for robust strategies to reinforce safety measures, advocate for healthier food options, and support mental health. Spearheading advancements in food quality detection, extensive education on safety protocols, and deep collaboration with leading health authorities are crucial steps toward achieving these objectives. Enforcing rigorous food safety measures involves the strict implementation of meticulous hygiene protocols, comprehensive training for food handlers, and systematic disinfection of surfaces to mitigate any potential contamination risks. Thoughtfully designed public awareness campaigns play a pivotal role in promoting well-balanced and nutritious diets, emphasizing the consumption of fresh, health-promoting produce, and nutrients known to positively influence mental well-being, such as omega-3 fatty acids and B vitamins (Table 1). In addressing emotional well-being during the pandemic, it is imperative to ensure the availability of robust mental health support services and the development of effective coping mechanisms for individuals experiencing emotional distress, anxiety, or depression. Extensive research and surveys have provided valuable insights into trends related to unhealthy food consumption and sedentary behaviors, leading to the formulation of targeted interventions that advocate for increased physical activity and healthier habits. Furthermore, advancements in food quality detection, including the implementation of cutting-edge omics techniques and revolutionary whole genome sequencing (WGS), have significantly enhanced the speed and accuracy of pathogen detection and tracing within food products. Rigorous adherence to health authority guidelines for food preparation and delivery, along with measures such as limited customer entry and the provision of essential sanitization supplies at store entrances, has strengthened safety measures across food service establishments. Ensuring a safe and high-quality food chain requires meticulous adherence to Good Farming Practices, which prioritize economic, environmental, and social sustainability in on-farm processes. Concurrently, extensive education on foodborne illnesses and the importance of adopting safe practices has been disseminated to food handlers and consumers alike. Strengthening national food systems and legislative frameworks has further bolstered food safety regulations. Employee education on food safety remains a top priority, with intensive training programs and certification initiatives targeting restaurant staff and food handlers. Collaborative efforts with esteemed health authorities, including government agencies and organizations such as the WHO, have been instrumental in devising and implementing effective food safety strategies during the pandemic. Active involvement in systemic disease prevention programs has been crucial in detecting and preempting public health risks related to food safety. Lastly, the dissemination of knowledge on food safety through educational seminars and informational materials has played a vital role in raising awareness and promoting conscious food choices, particularly during these challenging times when supporting overall health and mental well-being is of utmost importance.

Many papers highlighted the connection between the brain and the gut. The term “psychobiotics”, defined as the probiotic bacteria-derived molecules exerting the psychological potential to support mental health by targeting microbial interventions, was newly coined in 2013. The therapeutic potential of psychobiotics ranges from mood changes and anxiety to neurodegenerative diseases and neurodevelopmental disorders [125]. The influence of psychobiotics and fecal microbial transplantation (FMT) on the gut microbiome and behaviors related to mental health and the relationship between neurodevelopmental disorders and psychobiotics has received considerable attention in recent years. There is a bidirectional interaction in the microbiota–gut–brain (MGB) axis, and its modulation benefits brain activity and behavior as potential treatments [126]. Given these considerations, the gut–brain axis is an attractive target for the development of novel therapies for brain and mental health, such as probiotics [127]. Psychobiotics have the potential to not only reconstruct the gut barrier function by resisting harmful bacteria but also exert an immunomodulatory effect by reducing circulating hormones and pro-inflammatory cytokines in serum. The gut microbiota has been demonstrated to serve as a regulator of intestinal, systemic, and CNS resident immune cell function [128]. FMT is the transfer technique of a healthy donor’s fecal specimen to the GI tract of a recipient patient to reestablish the normal gut microbiome. This technique has been focused on in recent years because of the technical advances in metagenomics sequencing and the growing understanding of its function. FMT has been demonstrated to reconstruct a normally functioning microbial community, making it an accepted therapy with biological plausibility [129,130]. Researchers have shown the positive effects and potential pathways of promising therapeutic interventions, including psychobiotic supplementation and modulation of the gut.


foods-13-01610-t001_Table 1Table 1Nutritional elements and its influences on health: empowering defense against COVID-19.NutrientsRich SourcesEffectMechanism of ActionReferencesVitamin CCitrus fruits (e.g., oranges, lemons), strawberries, kiwi, and bell peppers
▪Acts as an antioxidant, potentially reducing inflammation.▪Enhanced immune function.

▪Vitamin C supports immune cell function and helps neutralize harmful reactive oxygen species, reducing oxidative stress and inflammation.▪Vitamin C influences cellular immunity and plays a role in the growth and differentiation of natural killer cells and T-cells. It enhances the function of neutrophils, promotes phagocytosis and supports the migration of neutrophils to the site of infection.▪Vitamin C has been found to reduce the likelihood of a cytokine storm, an excessive and harmful immune response, in some patients with COVID-19. This can help prevent severe inflammation and tissue damage.
[84,131,132]Vitamin DSun exposure, fatty fish (e.g., salmon, mackerel), and fortified dairy products
▪Enhancement of the innate immune system.▪regulation of blood pressure.

▪Vitamin D stimulates the production of antimicrobial peptides, such as cathelicidin LL-37, which have antimicrobial and antiviral properties. These peptides can disrupt the cell membranes of bacteria and viruses, including respiratory viruses like SARS-CoV-2.▪Vitamin D modulates the immune response, enhances antimicrobial peptide production and reduces excessive inflammation.▪Vitamin D has regulatory effects on the renin-angiotensin-aldosterone system (RAAS), which regulates blood pressure and fluid balance.
[133,134]ZincOysters, beef, pumpkin seeds, and chickpeas
▪Supports immune function and may have antiviral properties against coronaviruses.

▪Zinc helps in the replication and transcription of viral RNA, hindering viral proliferation.
[86,135] Omega-3 Fatty AcidsFatty fish (e.g., salmon, sardines), flaxseeds, and chia seeds
▪Managing the cytokine storm associated with severe COVID-19 cases.

▪Omega-3 fatty acids compete with pro-inflammatory fatty acids, leading to the production of less inflammatory molecules.
[136,137,138]QuercetinOnions, apples, and berries (e.g., blueberries, cranberries)
▪Exhibits antiviral properties and may reduce viral entry.▪Immunomodulatory activity.

▪Quercetin may interfere with viral cell entry, limiting viral replication.▪Quercetin has been shown to inhibit the expression of human ACE2 receptors, which are used by the SARS-CoV-2 virus to enter host cells.
[131,139,140]N-acetylcysteine (NAC)Garlic, onions, and cruciferous vegetables.
▪Boosts glutathione production, supporting antioxidant defenses and reducing inflammation.

▪NAC replenishes intracellular glutathione, neutralizing free radicals and modulating inflammatory pathways.
[141,142,143]SeleniumBrazil nuts, fish (e.g., tuna, halibut), and whole grains
▪Reducing oxidative stress.▪Anti-inflammatory properties.

▪Selenium is an essential component of antioxidant enzymes, such as glutathione peroxidase, which help reduce oxidative stress in the body.▪Declining expression of ACE-2 receptor: ACE-2 receptors are the entry point for SARS-CoV-2 into host cells. Selenium may help decrease the expression of these receptors, potentially reducing viral entry.
[144,145]Vitamin ASweet potatoes, carrots, and leafy greens (e.g., spinach, kale)
▪Supports the integrity of the respiratory epithelium and boosts the immune response.

▪Vitamin A is essential for maintaining mucosal surfaces, including the respiratory tract and enhances immune cell activity.
[146,147,148]Vitamin ENuts (e.g., almonds, sunflower seeds) and vegetable oils (e.g., sunflower oil and safflower oil)
▪An antioxidant that protects cells from oxidative stress.▪Immunomodulatory nutrient.

▪Vitamin E scavenges free radicals, reducing cellular damage caused by inflammation and oxidative stress.
[149,150,151]MagnesiumSpinach, nuts, seeds, and whole grains.
▪Supports immune function▪Acts as an antioxidant▪Bronchial smooth muscle relaxation.

▪Magnesium regulates immune responses and inhibits inflammatory signaling pathways.▪Magnesium can help relax bronchial smooth muscles, which may alleviate respiratory symptoms in COVID-19 patients.
[152,153]Vitamin B6Chicken, turkey, potatoes, and bananas.
▪Supports immune function.▪Supporting endothelial integrity.

▪Vitamin B6 is involved in immune cell maturation and antibody production.▪It can help relieve COVID-19 symptoms by improving immune responses, reducing pro-inflammatory cytokines, supporting endothelial integrity, and preventing hypercoagulability.
[154,155,156,157]ProteinMeats, poultry, fish, eggs, dairy products, legumes, nuts, and seeds
▪Maintaining and supporting the immune system, which is crucial in fighting viral infections like COVID-19.▪Protein is essential for tissue repair and wound healing.▪Improve respiratory muscle strength and function.

▪Protein aids in producing antibodies, cytokines, and other immune molecules that help fight off the virus.▪Certain proteins, such as the spike protein of the SARS-CoV-2 virus, are potential targets for drug development. Inhibiting the interaction between the spike protein and the human ACE2 receptor can prevent viral entry into host cells.
[154,155,158]CarbohydratesWhole grains, fruits, vegetables, legumes, and dairy products
▪The primary source of energy for the body, including immune cells.▪Support immune cell function and improve immune outcomes.▪Reduce inflammation and attenuate metabolic disorders.▪Managing metabolic disorders associated with COVID-19.

▪Certain types of carbohydrates, like resistant starch, can promote beneficial bacterial growth in the gut, leading to increased production of short-chain fatty acids (SCFAs) with anti-inflammatory properties.▪Consuming carbohydrates with a lower glycemic index can help regulate blood glucose levels and reduce the risk of insulin resistance.
[157,159]


## 5. Conclusions

The COVID-19 pandemic has presented unparalleled obstacles to global health and well-being, affecting every facet of life, including food safety and mental health. Although distinct from influenza, the SARS-CoV-2 virus shares similarities and spreads through diverse transmission routes, necessitating stringent isolation measures. As a result, the food industry has encountered significant disruptions, leading to noticeable shifts in food consumption patterns during quarantine periods. While some individuals have made strides toward healthier dietary choices, others have struggled to maintain optimal eating habits. The collaboration between food advisory services and COVID-19 response efforts underscores the crucial role of communication and guidance in safeguarding public health during crises. Moreover, the pandemic has illuminated the intrinsic link between food choices and mental health, emphasizing the imperative of promoting healthier dietary behaviors. Additionally, disruptions in food production and supply chains have underscored the necessity for robust food safety measures and advancements in food quality detection technologies. This study contributes to the existing body of knowledge by uniquely linking food security, safety, and psychological wellness within the context of a global health crisis. Our integrated approach not only highlights the immediate impacts of COVID-19 on food systems and mental health but also offers sustainable strategies for future resilience. In addition, by addressing the intertwined nature of these issues, our findings provide valuable insights and practical recommendations for enhancing food security and psychological well-being, thus supporting both academic research and real-world applications in crisis management and policy development.

## Figures and Tables

**Figure 1 foods-13-01610-f001:**
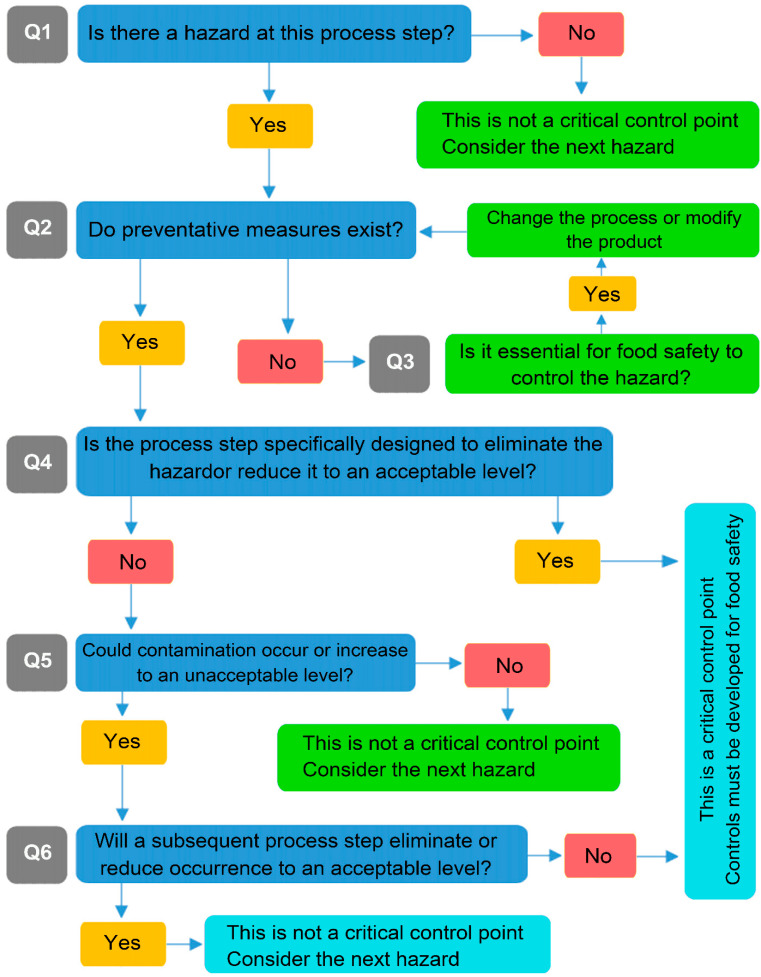
Derived decision tree for food safety [19,20].

**Figure 2 foods-13-01610-f002:**
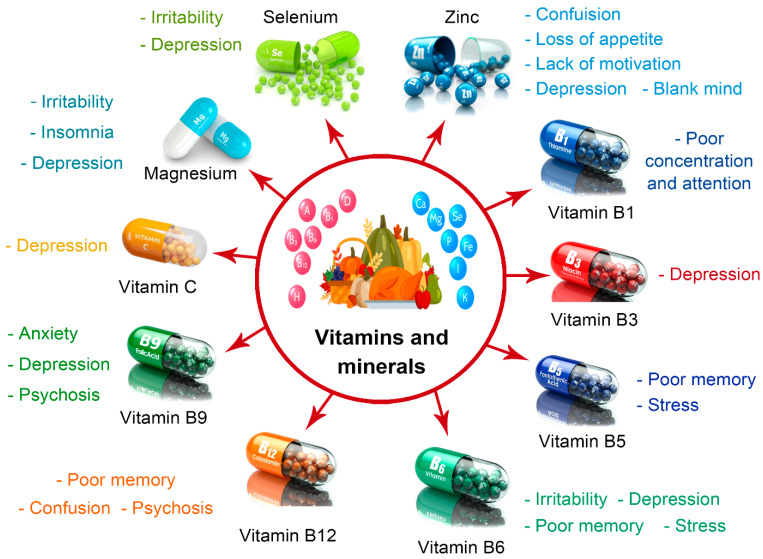
Nutrient deficiency and its related disorders.

**Figure 3 foods-13-01610-f003:**
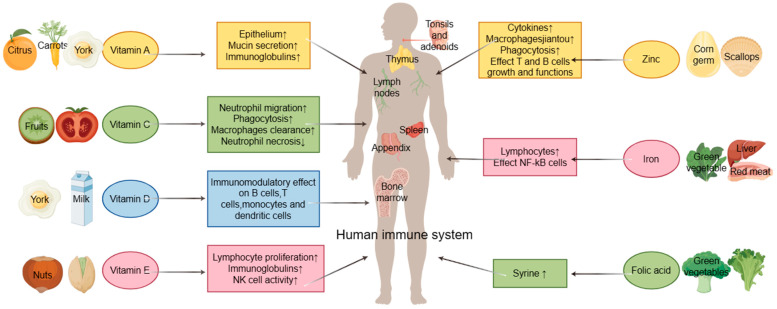
Boosting mechanisms on the immune system and representative food sources of critical micronutrients (By Figdraw; https://www.figdraw.com/#/; accessed on 23 January 2024).

**Figure 4 foods-13-01610-f004:**
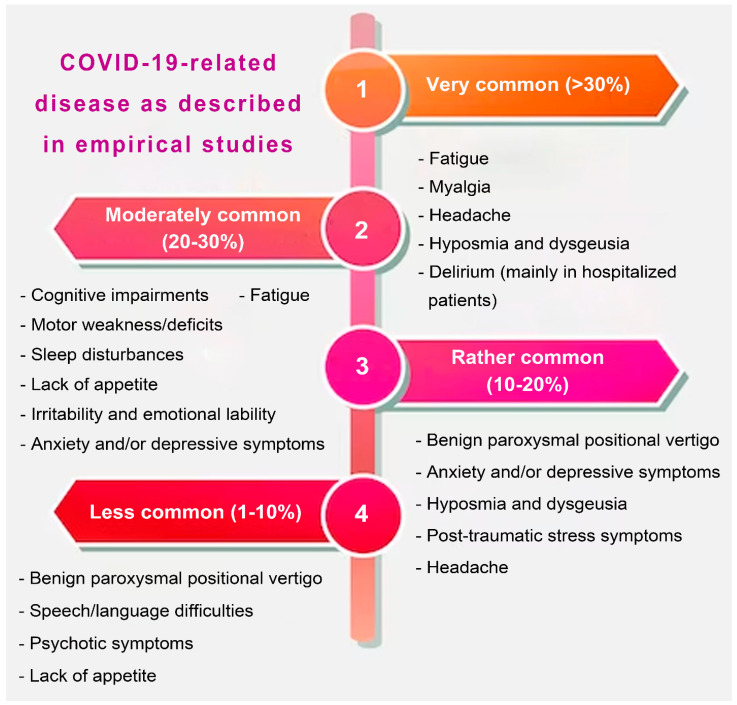
COVID-19-related disease as described in empirical studies.

**Figure 5 foods-13-01610-f005:**
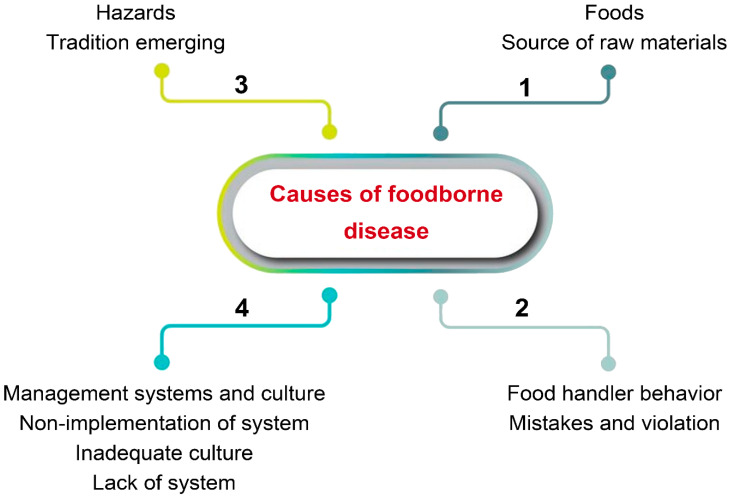
Flowchart of significant causes of foodborne disease.

**Figure 6 foods-13-01610-f006:**
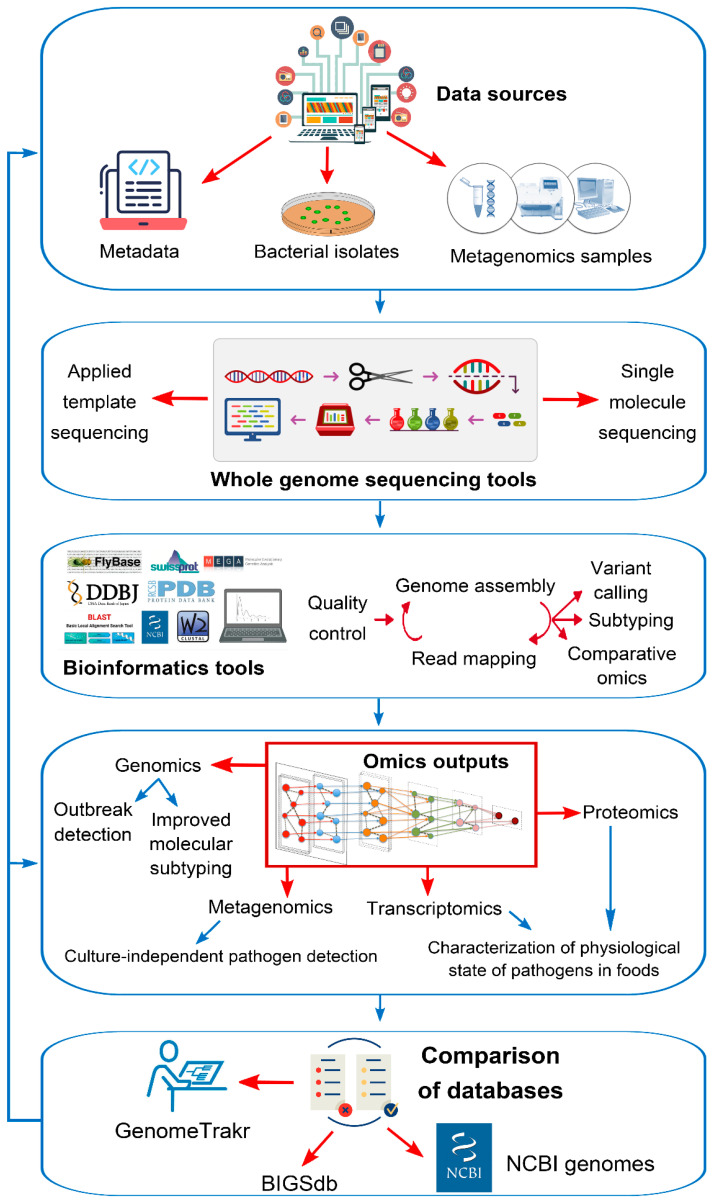
Flow diagram illustrating current food safety omics techniques, focusing on whole genome sequencing of bacterial foodborne pathogens for enhanced surveillance.

## Data Availability

No new data were created or analyzed in this study. Data sharing is not applicable to this article.

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
