# Peer review of "Exploring the Interplay of Food Security, Safety, and Psychological Wellness in the COVID-19 Era: Managing Strategies for Resilience and Adaptation"

_foods, 2024, doi:10.3390/foods13111610_

Round 1

Reviewer 1 Report

Comments and Suggestions for Authors

Fanrui Zhou et al. submitted to Foods a review, focusing on the psychological well-being in the COVID-19 pandemic, dealing with food safety and food security.

This manuscript presents many inaccuracies and, moreover, it is a series of paragraphs disconnected from each other from the point of view of the logic of the review. The entire structure of the manuscript must certainly be revised, following a logical thread. The article is very difficult to read and follow in context.

It is essential to specifically explain how the review was conceived, created and structured and according to which inclusion criteria.

Regarding the sentence “The COVID-19 outbreak was declared a pandemic by the World Health Organization (WHO) in 2019”: please check this sentence, the right year should be March 2020.

Line 117: Reference is made to Figure 1, but should be Figure 2.

Furthermore, consider that the title refers to “Managing Strategies for Resilience and Adaptation”, explain specifically what these strategies are in the discussion section, in addition to the take-home messages that derive from your work.

Comments on the Quality of English Language

Moderate editing of English language required

Author Response

The authors are very much thankful to the anonymous reviewer for careful review and constructive critical comments on our manuscript entitled “Exploring the Interplay of Food Security, Safety, and Psychological Wellness in the COVID-19 Era: Managing Strategies for Resilience and Adaptation” (food-2949864). The manuscript has been revised as per suggestions and answers are given below in detail. We are confident that academic quality of the revised version has been improved now and it will be accepted for the publication in its present form.

Best regards and appreciations,

Prof. Imran Haider Shamsi

Corresponding author.

Reviewer 1

Fanrui Zhou et al. submitted to Foods a review, focusing on the psychological well-being in the COVID-19 pandemic, dealing with food safety and food security.

This manuscript presents many inaccuracies and, moreover, it is a series of paragraphs disconnected from each other from the point of view of the logic of the review. The entire structure of the manuscript must certainly be revised, following a logical thread. The article is very difficult to read and follow in context.

Response: Thank you for taking the time to review our manuscript and for your insightful feedback. We sincerely apologize for any inaccuracies and lack of logical coherence in the structure of the manuscript that may have hindered its readability and understanding for the respected anonymous reviewer. We understand the importance of presenting information in a clear and organized manner to facilitate comprehension for readers so we have carefully revised the entire structure of the manuscript to ensure that it follows a logical thread and presents information in a cohesive manner.

It is essential to specifically explain how the review was conceived, created and structured and according to which inclusion criteria.

Response: We appreciate the reviewer's insightful comment regarding the conception, creation and structure of our review. In response, we have provided a detailed explanation of our methodology and inclusion criteria in revised manuscript (Line 77-89).

Regarding the sentence “The COVID-19 outbreak was declared a pandemic by the World Health Organization (WHO) in 2019”: please check this sentence, the right year should be March 2020.

Response: Thank you for pointing out this mistake and correction has been made (Line 40).

Line 117: Reference is made to Figure 1, but should be Figure 2.

Response: We would like to acknowledge you for pointing this. Reference has been corrected in revised manuscript (Line 131).

Furthermore, consider that the title refers to “Managing Strategies for Resilience and Adaptation”, explain specifically what these strategies are in the discussion section, in addition to the take-home messages that derive from your work.

Response: Thank you again for providing detailed feedback on the discussion section of our article. We appreciate your insights and suggestions for improvement. We have carefully reviewed and revised the discussion section to address the issues you've highlighted (Line 504-523).

Thank you for your valuable time and professional guidance.

Reviewer 2 Report

Comments and Suggestions for Authors

Dear Authors

You have tried to summarise answers to complex but essential questions about food supply security and food safety. They did it on the basis of 148 articles, which means that this number is quite a bit low. And there is not a single reference in section 3.3. and in 4.3.

In section 3.4 which is general there is no relationship to COVID-19.

Sections 3.3 and 3.5 are too general and should be more specific.

Part 3. 2. is also a very good compilation, but lacks the link to food and COVID-19.

I think the statement in lines 123 and 124 should be refined with the level of development difference.
The Summary should also be changed because there are few forward looking findings. Overall good, but needs to be more in line with the title.

You had not planned to present one or two of the food chains that have been greatly affected by COVID-19?
What are your views on the relationship between food advisory and COVID-19? 

Comments on the Quality of English Language

Please use more the words food chain, food security and food safety.

Author Response

The authors are very much thankful to the anonymous reviewer for careful review of our manuscript entitled “Exploring the Interplay of Food Security, Safety, and Psychological Wellness in the COVID-19 Era: Managing Strategies for Resilience and Adaptation” (food-2949864). The manuscript has been revised as per suggestions and answers are given below in detail. We are confident that academic quality of the revised version has been improved now and it will be considered for publication in its present form.

Warm regards and appreciations,

Prof. Imran Haider Shamsi

Reviewer 2

Dear Authors

You have tried to summarise answers to complex but essential questions about food supply security and food safety. They did it on the basis of 148 articles, which means that this number is quite a bit low. And there is not a single reference in section 3.3. and in 4.3.

Response: Thank you very much for taking the time to review our article and we appreciate your feedback and constructive comments. For sure your comments have helped us alot to improve the quality of the manuscript. Regarding your comment about the number of references cited, While we acknowledge that this may seem limited in the scope of such a complex topic, we carefully selected these references to encompass a comprehensive range of perspectives and findings in the field, there was a choice of more than 200 though but it could add a big bulk. Our clear intention was to provide a focused yet inclusive examination of the subject matter, drawing on the key studies to support our arguments effectively. For section 3.3 (now 3.4) and 4.3 (now 4.4) after modification and required changes, we felt that citing additional sources would not significantly enhance the clarity or depth of the discussion at hand. Instead, we focused on presenting our findings and interpretations in a coherent and logical manner, guided by the overarching objectives of the study. We do expect with respect that you may agree with our point of view now. Thanks again.

In section 3.4 which is general there is no relationship to COVID-19.

Response: Thank you for your insightful comments regarding section 3.4 (now 3.5) of our article. While we appreciate your suggestion, it's important to note that section 3.5 of our article primarily focuses on the influence of nutrition and hunger on mental health, particularly highlighting the physiological and psychological mechanisms through which food and hunger impact mood and emotions. Given the specific focus of this section, we chose to delve into the intricate relationship between dietary factors, brain chemistry and emotional regulation, as supported by the literature cited. We did rewrite the section as well.

Sections 3.3 and 3.5 are too general and should be more specific.

Response: Thank you for your thoughtful feedback regarding Sections 3.3 (now 3.4) and 3.5 (now 3.6) of our article. For these two sections, we rewrote and intended to maintain the current structure while making slight adjustments to offer more targeted insights into the relationship between mental health and food preferences. Our approach has involved refining the discussion to highlight specific research findings and psychological mechanisms underlying emotional eating behaviors and food choices. Hopefully you will kindly accept our justification, thank you.

Part 3. 2. is also a very good compilation, but lacks the link to food and COVID-19.

Response: Thank you for your feedback regarding Part 3.2 of the manuscript. We appreciate your acknowledgment of the quality of the compilation and your suggestion to include a link to food and COVID-19 within this section. In response, we have revised section 3.2 to incorporate relevant information.

I think the statement in lines 123 and 124 should be refined with the level of development difference.

Response: Thank you for your feedback regarding lines 123 and 124 (now 140-142) of the manuscript. We have revised the line to make it clearer and more precise in the revisions.

The Summary should also be changed because there are few forward looking findings. Overall good, but needs to be more in line with the title.

Response: Thank you for your feedback regarding the Summary of our manuscript. In response, we have revised this section to include more forward-looking findings that align closely with the title of the article (Line 536-558). Please accept, thanks.

You had not planned to present one or two of the food chains that have been greatly affected by COVID-19?

Response: Thank you for your insightful comment regarding the impact of COVID-19 on food chains. We appreciate your suggestion and added section 4.1 in the revised manuscript to incorporate relevant information on big data impacting dynamic food safety in the food chains affected by COVID-19. We believe that this addition has enhanced the relevance and timeliness of the manuscript, providing valuable insights into the dynamic relationship between COVID-19 and food chains. Thanks indeed.

What are your views on the relationship between food advisory and COVID-19?

Response: Thank you so much for drawing our attention towards this important aspect. We understood the significance of this point so relevant information has been added into introduction (line 46-51) and conclusion (line 544-546) sections, hopefully you will accept.

Reviewer 3 Report

Comments and Suggestions for Authors

The review article “Exploring the Interplay of Food Security, Safety, and Psychological Wellness in the COVID-19 Era: Managing Strategies for Resilience and Adaptation” provides a comprehensive overview of the intersection between the COVID-19 pandemic, mental health, and food consumption, discussing various aspects ranging from the societal impacts of the pandemic to the physiological effects of nutrients on mental well-being.

The review is concise, well-written, and presents valuable information on the complex interplay between food consumption, food safety, mental health, and COVID-19. The manuscript can be accepted after minor revision.

- It would be worthwhile to make a brief discussion on the stressors that affect mental health during the COVID-19 pandemic.

- I suggest adding a section regarding the role of dietary nutrients on the immune system in the COVID-19 context, highlighting the influence of nutrients and bioactive molecules present in foodstuffs on the nutritional status of patients and the dietary recommendations for COVID-19.

- Within the section exploring the impact of nutrients on mental well-being, there is a sudden mention of nano-curcumin and its effects on cytokine levels. However, this information seems to be somewhat disconnected from the broader discussion and could be better integrated or explained within the context of the article. It is recommended that the content be reviewed and revised for cohesiveness and clarity.

- I recommend adding a paragraph about the psychobiotic products and their impact on mental health.

Author Response

The authors are very much thankful to the anonymous reviewer for the careful review and constructive critical comments on our manuscript (food-2949864). The manuscript has been revised as per suggestions and answers are given below in detail. We are confident that academic quality of the revised version has been improved now and it will be accepted for the publication in its present form.

Best regards and appreciations,

Prof. Imran Haider Shamsi

Reviewer 3

The review article “Exploring the Interplay of Food Security, Safety, and Psychological Wellness in the COVID-19 Era: Managing Strategies for Resilience and Adaptation” provides a comprehensive overview of the intersection between the COVID-19 pandemic, mental health, and food consumption, discussing various aspects ranging from the societal impacts of the pandemic to the physiological effects of nutrients on mental well-being.

The review is concise, well-written, and presents valuable information on the complex interplay between food consumption, food safety, mental health, and COVID-19. The manuscript can be accepted after minor revision.

Response: Dear Reviewer, thank you very much indeed for your encouraging comments and for your detailed review and constructive feedback on our manuscript. We appreciate the opportunity to improve the quality and clarity of our work. We believe these revisions have significantly improved the manuscript's quality, making it more informative and beneficial to the readers. We hope these modifications meet your expectations. Following are the detailed answers for the queries:

- It would be worthwhile to make a brief discussion on the stressors that affect mental health during the COVID-19 pandemic.

Response: We would like to acknowledge you for pointing this. We have included a brief discussion about stressors that affect mental health during the COVID-19 pandemic in Introduction section (Line 52-65). Thanks.

- I suggest adding a section regarding the role of dietary nutrients on the immune system in the COVID-19 context, highlighting the influence of nutrients and bioactive molecules present in foodstuffs on the nutritional status of patients and the dietary recommendations for COVID-19.

Response: We appreciate the reviewer's insightful comment regarding addition of the role of dietary nutrients on the immune system in the COVID-19 context. We have added relevant information into the revised manuscript under heading “3.3 Impact of Nutrients on Immune System” (Line 220-257). Thank you.

- Within the section exploring the impact of nutrients on mental well-being, there is a sudden mention of nano-curcumin and its effects on cytokine levels. However, this information seems to be somewhat disconnected from the broader discussion and could be better integrated or explained within the context of the article. It is recommended that the content be reviewed and revised for cohesiveness and clarity.

Response: We appreciate your insightful comments again regarding the integration of information about nano-curcumin within the broader discussion on the impact of nutrients on mental well-being. We also understand your concern about the sudden mention of nano-curcumin and its effects on cytokine levels. Our intention in including this information was to highlight emerging research on the potential role of dietary supplements in promoting mental well-being through their modulation of inflammatory responses in the body. To address this concern, we have revised the manuscript to include transitional lines (210-217) that provides context for the mention of nano-curcumin within the broader framework of the article. Thank you.

- I recommend adding a paragraph about the psychobiotic products and their impact on mental health.

Response: We appreciate the reviewer's insightful comment again and totally agreed with the professional approach, regarding the importance of discussing psychobiotics products and their impact on mental health. In response, we have added a paragraph regarding this in revised manuscript (Line 475-500). Thanks a lot once again.

Round 2

Reviewer 1 Report

Comments and Suggestions for Authors

The Authors have implemented this manuscript, which however does not at all take on the typical features of a review. In fact, it remains a treaty that does not provide substantial added value to the numerous data already present in biomedical literature.

Comments on the Quality of English Language

Minor editing of English language required

Author Response

Reviewer's comments: The Authors have implemented this manuscript, which however does not at all take on the typical features of a review. In fact, it remains a treaty that does not provide substantial added value to the numerous data already present in biomedical literature.

Authors' response: Thank you again for your feedback on our manuscript. We appreciate the opportunity to address your concerns regarding the nature of our review paper. However, we would like to clearly state that the aim of this study is to provide a comprehensive examination of the intricate relationships between food security, safety and psychological wellness specifically within the context of the pandemic. Furthermore, this study aimed to contribute to a deeper understanding of the complex dynamics and offers practical strategies for resilience and adaptation in these challenging times. Nevertheless, we have carefully considered your feedbacks/comments of round 1 and have incorporated all your suggestions ensuring that the rewritten revised version of the manuscript, which further clarifies your suggestions. Thank you again for your valuable input.

Reviewer 2 Report

Comments and Suggestions for Authors

Perhaps the keywords Food loss and waste should be replaced because there is not so much mention of them. It might even be good if it is not the first initial term here.

Author Response

Reviewer's comment: Perhaps the keywords Food loss and waste should be replaced because there is not so much mention of them. It might even be good if it is not the first initial term here.

Authors' response: Thank you very much for your valuable feedback regarding the choice of keywords in our manuscript. Suggested amendment has been made in the revised manuscript. Thanks again for your professional guidance.